# Characterization of the Dual Functions of *LvCrustinVII* from *Litopenaeus vannamei* as Antimicrobial Peptide and Opsonin

**DOI:** 10.3390/md20030157

**Published:** 2022-02-22

**Authors:** Jie Hu, Shihao Li, Qian Lv, Miao Miao, Xuechun Li, Fuhua Li

**Affiliations:** 1School of Marine Science and Engineering, Qingdao Agricultural University, Qingdao 266109, China; hujie@qdio.ac.cn (J.H.); lvqian@qdio.ac.cn (Q.L.); 2CAS and Shandong Province Key Laboratory of Experimental Marine Biology, Institute of Oceanology, Chinese Academy of Sciences, Qingdao 266071, China; miaomiao@qdio.ac.cn (M.M.); lixuechun17@mails.ucas.ac.cn (X.L.); fhli@qdio.ac.cn (F.L.); 3Laboratory for Marine Biology and Biotechnology, Qingdao National Laboratory for Marine Science and Technology, Qingdao 266237, China; 4Center for Ocean Mega-Science, Chinese Academy of Sciences, Qingdao 266071, China; 5University of Chinese Academy of Sciences, Beijing 100049, China; 6The Innovation of Seed Design, Chinese Academy of Sciences, Wuhan 430072, China

**Keywords:** antimicrobial peptides (AMPs), crustin, antibacterial activity, opsonin, *Litopenaeus vannamei*

## Abstract

Crustin are a family of antimicrobial peptides that play an important role in protecting against pathogens infection in the innate immune system of crustaceans. Previously, we identified several novel types of crustins, including type VI and type VII crustins. However, their immune functions were still unclear. In the present study, the immune function of type VII crustin LvCrustinVII were investigated in *Litopenaeus vannamei*. *LvCrustinVII* was wildly expressed in all tested tissues, with relatively high expression levels in hepatopancreas, epidermis and lymphoid organ. Upon *Vibrio parahaemolyticus* infection, *LvCrustinVII* was significantly upregulated in hepatopancreas. Recombinant LvCrustinVII (rLvCrustinVII) showed strong inhibitory activities against Gram-negative bacteria *Vibrio harveyi* and *V. parahaemolyticus*, while weak activities against the Gram-positive bacteria *Staphylococcus aureus*. Binding assay showed that rLvCrustinVII could bind strongly to *V. harveyi* and *V. parahaemolyticus*, as well as the cell wall components Glu, LPS and PGN. In the presence of Ca^2+^, rLvCrustinVII could agglutinate *V. parahaemolyticus* and enhance hemocyte phagocytosis. The present data partially illustrate the immune function of *LvCrustinVII*, which enrich our understanding on the functional mechanisms of crustins and provide useful information for application of this kind of antimicrobial peptides.

## 1. Introduction

In recent years, aquaculture develops rapidly in China, but it faces serious problems with disease outbreaks. Many antibiotics are used to deal with this problem [1]. However, the widespread use of antibiotics leads to the production of many drug-resistant microorganisms, which often causes more serious infection and morbidity [2]. Due to the failure of most conventional antibiotics to counteract “superbugs” and the dwindling supply of new ones, there is an urgent need to develop other antibacterial drugs [3,4]. Antimicrobial peptides (AMPs) provide a possible solution to the antibiotic resistance crisis [5]. Compared with traditional antibiotics, AMPs have the advantages of less resistance, least toxicity to the host, broad spectrum activity and rapid killing ability [6,7]. These properties make them the best alternative to fight against bacteria [8]. Over the years, many research groups have focused on AMPs and greatly improved our understanding of how AMPs exert their antibacterial effects [9,10].

AMPs are a class of small molecules that can help the body resist external pathogens, and kill a variety of bacteria, viruses, parasites, etc. [11,12,13]. They are considered as the body’s first line of defense against invading microorganisms. AMPs are usually secreted by blood cells, as well as epithelial cells in tissues, skin, etc., and interact directly with many bacteria in contact with the body [14]. In addition, various cells of the immune system, especially phagocytes, also produce AMPs and store them in cellular granules [15]. In response to infection, these phagocytes release them into hemolymph and directly kill pathogens through targeting and destroying microbial cell membranes [16].

AMPs widely exist in nature, and they are an important part of innate immunity in various organisms. AMPs produced by the innate immunity in crustaceans present some novel drug properties. As reviewed by Matos and Rosa [17], ALF-derived peptides are diverse bioactive molecules, modulating inflammatory responses and displaying activity against human parasites, viruses and tumors. Some PENs can behave as cytokines attracting hemocytes towards sites of injury and bind to chitin, suggesting a role in wound healing. Hyastatins and Arasins could participate wound healing and/or molting process. Distinct crustin members have been shown to promote phagocytosis, bacteria agglutination and haematopoiesis. Over the past few decades, different types of AMPs have been found in crustaceans, including penaeidins, anti-lipopolysaccharide factors (ALFs), lysozymes, and crustins [18,19,20,21,22,23,24,25]. Crustins are a kind of antibacterial peptides secreted mainly by crustaceans [16]. They are featured with a C-terminal whey acidic protein (WAP) domain, which consists of about 50 amino acids and eight conserved cysteine residues that form a characteristic four-disulfide bonds (4DSC) [26]. The WAP domain is a key structural basis for its biological activity [27]. In addition, the amino acid polytropic region located between the signal peptide sequence and the WAP domain is important for its functions [28]. Crustins are classified into seven types mainly based on their amino-acid composition features of the polytropic region [29,30]. Different types of crustins exhibit distinct biological functions. Type I crustins show antibacterial activity mainly against Gram-positive bacteria [30]. Type II crustins present antibacterial activity against both Gram-positive and Gram-negative bacteria [30]. Type I and II crustins were also reported to have regulatory functions on host microbial communities [16,26,31,32,33]. Type III and IV crustins have protease inhibitory activity and some exhibit antimicrobial activity [34,35]. Type V crustins only exist in ants but are absent in crustaceans [36]. Type VI and VII crustins are newly reported types found in shrimp [29]. Type VI crustin consists of a signal peptide, a glycine-rich region and one WAP domain, which lacks the cysteine-rich region compared to type II crustins. Type VII crustin consists of a signal peptide, a serine/threonine-rich region, a cysteine-rich region and one WAP domain. However, the antimicrobial activities of these types of crustins are still less investigated.

Clarifying the activity characteristics of newly identified AMPs is the basis for drug development and utilization. In the present study, we focus on the antimicrobial activity of one type VII crustin, LvCrustinVII, identified in *L. vannamei*. It was widely expressed in tested tissues and apparently responsive to *Vibrio parahaemolyticus* infection. The recombinant LvCrustinVII protein showed antibacterial activity, microorganisms binding activity and polysaccharides binding activity. In addition, LvCrustinVII could also agglutinate *V. parahaemolyticus* and greatly enhance phagocytosis of shrimp hemocytes in the presence of Ca^2+^. These results suggest that LvCrustinVII might exert dual functions as AMP and opsonin in host immune system to fight against pathogens infection.

## 2. Results

### 2.1. Tissue Distribution and Immune Responses of LvCrustinVII Transcripts

The expression levels of *LvCrustin VII* transcripts were detected in seven tissues of shrimp by qRT-PCR. The results showed that *LvCrustinVII* transcripts were widely distributed in all tested tissues, with relatively higher expression levels in epidermis, hepatopancreas and lymphoid organ, followed by intestine, stomach, hemocytes and gill (Figure 1A).

In order to know whether it participates in host immune defense, the immune responses of *LvCrustinVII* were detected in shrimp after *Vibrio parahaemolyticus* infection. In hepatopancreas, the expression level of *LvCrustinVII* was significantly up-regulated at 24 hpi after *V. parahaemolyticus* infection, which was 9.03-fold compared to the control group (Figure 1B). The differential expression of *LvCrustinVII* suggests that it is involved in immune defense against *Vibrio* infection.

### 2.2. Expression and Purification of the Recombinant LvCrustinVII

The recombinant LvCrustinVII (rLvCrustinVII) was produced in *E. coli BL21(DE3)* by constructing the expression plasmid pET-28a-*LvCrustinVII*. After IPTG induction, the recombinant protein was expressed at 2 h (Figure 2A). The whole cell lysate was analyzed by SDS-PAGE, and the results showed that rLvCrustinVII existed in the inclusion body (Figure 2B). The recombinant protein was purified under denaturing condition using a TALON Resins affinity chromatography and then refolded to native state, and the recombinant protein was collected after concentration. The recombinant protein *LvCrustinVII* had a distinct band with molecular mass of about 24.24kD, which was consistent with the molecular mass predicted by the ExPASy (https://web.expasy.org/protparam/ (accessed on 15 December 2021)) (Figure 2C).

### 2.3. Antimicrobial Activity of rLvCrustinVII

Minimal inhibitory concentration (MIC) assay was used to test the antimicrobial activity of rLvCrustinVII against the Gram-positive bacteria *S. aureus* and Gram-negative bacteria *E. coli*, *V. harveyi* and *V. parahaemolyticus*. The results showed that rLvCrustinVII showed high antimicrobial activity against *V. harveyi* and *V. parahaemolyticus*, both with an MIC of 2.5 μM. On the contrary, the inhibitory activity of rLvCrustinVII against *S. aureus* and *E. coli* was low, with MICs more than 20 μM (Table 1).

### 2.4. Microorganism and Polysaccharides Binding Activity of rLvCrustinVII

To further learn the antibacterial mechanism of rLvCrustinVII, the above four bacteria were also used for microorganism binding assay. After incubation with Gram-negative bacteria *E. coli*, *V. harveyi* or *V. parahaemolyticus* at the final concentration of 2.5 µM, the rLvCrustinVII protein was mainly present in the eluted fractions, whereas slight bands were in the supernatant. After incubation with the Gram-positive bacteria *S. aureus*, the protein was mainly detected in the supernatant whereas slight bands were in the eluted fractions (Figure 3A). Western blot analysis also confirmed the results (Figure 3B). These results showed that rLvCrustinVII could strongly bind to three Gram-negative bacteria and weakly bind to *S. aureus*.

Since rLvCrustinVII can bind to bacteria, its binding properties to cell surface components, including β-1,3-Glucan (Glu), lipopolysaccharides (LPS) and peptidoglycan (PGN), were further analyzed by ELISA. The results showed that rLvCrustinVII had strong binding activity with Glu, LPS and PGN, and the binding ability increased gradually with the increase of rLvCrustinVII concentration (Figure 4). These results suggest that the microbial binding activity of rLvCrustinVII might depend on its binding activity to these bacterial polysaccharides.

### 2.5. Agglutinating and Phagocytosis-Enhancing Activitiy of rLvCrustinVII

After detecting the binding activity of rLvCrustinVII to microorganisms and microbial polysaccharides, the bacterial agglutination activity was further checked. The results showed that rLvCrustinVII had agglutination ability against *V. parahaemolyticus* (Figure 5A), and Ca^2+^ could greatly improve the agglutination ability of rLvCrustinVII (Figure 5B). No significant change in agglutination activity was observed in the negative controls (Figure 5C,D) and the blank controls (Figure 5E,F).

To investigate whether the agglutination ability of rLvCrustinVII could enhance the phagocytosis activity of host hemocytes, an in vitro phagocytosis assay was carried out. The results showed that the phagocytic rate of shrimp hemocytes incubated with rLvCrustinVII was 38.13% in the presence of Ca^2+^, which was much higher than that of hemocytes incubated with rLvCrustinVII in the absence of Ca^2+^ (24.67%) and those from control groups (16.73%–22.27%, Figure 6). The data suggest that rLvCrustinVII could greatly promote phagocytosis of shrimp hemocytes against *V. parahaemolyticus* in the presence of Ca^2+^.

## 3. Discussion

Crustin is an antibacterial peptide rich in cationic cysteine secreted by crustaceans [16]. The first crustin was identified as a natural protein in *Carcinus maenas* blood cells, named “carcinin”, with antimicrobial activity against Gram-positive bacteria [37]. Since then, different crustins have been found in shrimp, crayfish and crab [38,39,40]. The crustins can be divided into seven different types based on their amino acid sequences [29,41,42]. Type I crustins contain a signal peptide, followed by a cysteine-rich region and a WAP domain. Compared with type I crustins, type II crustins have an extra glycine-rich region between the cysteine-rich region and the WAP domain. Type III crustins, also known as single WAP domain containing peptides (SWDs), consist of only a short N-terminal region between the signal peptide and WAP domain. Type IV crustins contain two WAP domains, also known as double WAP domain containing peptides (DWDs). Type V crustins, which are only reported in insects, contain the signal peptide, cysteine-rich region, aromatic amino acids-rich region and WAP domain. Type VI and VII crustins are recently reported types found in shrimp [29]. Different from other types, the type VII crustin LvCrustinVII contains a serine/leucine-rich region between the cysteine-rich region and the signal peptide. Different amino acids composition might enable LvCrustinVII to have distinct antimicrobial activities.

As one kind of antimicrobial peptide, crustins are important components for the first line of host immune defense system. Most reported crustin genes are responsive to immune challenges of various pathogens [16,27,28,32,43]. Here, the widely distributed type VII crustin *LvCrustinVII* was responsive to *V. parahaemolyticus* infection in the hepatopancreas of *L. vannamei*. Moreover, the in vitro antimicrobial activity of LvCrustinVII against *V. parahaemolyticus* and *V. harveyi* rather than other bacteria suggests that LvCrustinVII is important in host defense against *Vibrio* infection. Moreover, the type VII crustin LvCrustinVII exhibits a different antimicrobial activity from previously reported types of crustins. Type I, II and III crustins mainly have strong antimicrobial activity against Gram-positive bacteria, and only a few type IIb and III crustins show strong antimicrobial activity against Gram-negative bacteria [30,44,45]. Most type IV crustins lack antimicrobial activity, with a few exceptions [45,46,47]. Unlike these types of crustins, the type VII crustin LvCrustinVII has strong inhibitory activity against *Vibrio* bacteria while weak activity against other tested bacteria, which would provide a new target for drug development against *Vibrio* infection.

Direct contact with microorganisms is the basis of antimicrobial peptides exerting their activities. Many antimicrobial peptides, such as anti-lipopolysaccharide factors and crustins, are reported as having direct binding ability with pathogens [43,48,49]. LvCrustinVII could strongly bind to *V. parahaemolyticus* and *V. harveyi*, suggesting that its antibacterial activity against *Vibrio* is closely related to its microorganism binding activity. Antimicrobial peptides usually have a positive electrical and amphiphilic structure, which enables them easy to interact with the negatively charged cell walls and phospholipid bilayers of pathogenic bacteria [43,45,50,51]. Microbial polysaccharides are main targets of antimicrobial peptides [28,46,48]. LvCrustinVII exhibits strong binding ability with microbial polysaccharides LPS, PGN and Glu with a concentration dependent manner. This indicates that the microorganism binding activity of LvCrustinVII largely relies on its binding to microbial polysaccharides. Although LvCrustinVII could strongly bind to *E. coli*, the inhibitory activity against it is not obvious. In *Scylla paramamosain*, the crustin SpCrus6 also showed low inhibitory activity against *Vibrio alginolyticus*, *V. parahemolyticus* and *V. harveyi* though it could strongly bind to them [28]. Therefore, the microorganism binding activity of LvCrustinVII is the necessary but not the sufficient condition for its antibacterial activity.

The direct contact also enables agglutinating functions with microorganisms of crustins. Some crustins display considerable bacterial agglutination activity by crosslinking the surface components of bacterial cells and forming lattices. The bacterial agglutination activity was considered as one of the main modes for crustin to exert its antibacterial activity [28]. In *Marsupenaeus japonicus*, injection of the crustin *Mj*Cru I-1 could increase the hemocyte phagocytosis against *V. anguillarum* and *S. aureus* infection [52]. Knockdown of the *crustin*-like gene in *M. japonicus* could decrease the phagocytic rate of shrimp hemocytes on WSSV [53]. These data suggest that crustins have positive regulatory functions on hemocytes phagocytosis. In the present study, LvCrustinVII could agglutinate *V. parahaemolyticus* and promote phagocytosis of shrimp hemocytes. Based on the above evidence, we consider that the agglutination activity of crustin is essential for its phagocytosis promoting function, which makes it function as an opsonin during pathogens infection. Interestingly, the agglutination activity and phagocytosis promoting function of LvCrustinVII could be largely enhanced in the presence of Ca^2+^. A similar phenomenon was also reported in *Scylla paramamosain*, where the agglutination activity of the crustin SpCrus6 to *B. megaterium* could be greatly strengthened in the presence of Ca^2+^ [28]. However, the underlying molecular mechanism needs to be further investigated. Collectively, the newly reported type VII crustin, LvCrustinVII exhibits dual functions as the antimicrobial peptide and opsonin in the innate immunity of shrimp.

## 4. Materials and Methods

### 4.1. Animals and Tissues Preparation

All the healthy shrimp *Litopenaeus vannamei* were reared in our lab for two months before the experiment, with an average length of 6.56 ± 0.54 cm and body weight of 3.42 ± 0.70 g. Different tissues including epidermis, gills, hemocyte cells, hepatopancreas, intestinal, lymphoid organ (Oka) and stomach were collected from nine individuals. Each tissue from three individuals was mixed as one sample and three replicates were prepared for each tissue. Hemolymph was obtained from the base of the fifth pereiopod and mixed with equal volume of anticoagulant (336 mM NaCl, 27 mM trisodium citrate, 9 mM EDTA-2 Na, 115 mM glucose). The hemocytes was collected by centrifugation at 800× *g*, 4 °C for 10 min.

The pathogenic *Vibrio parahaemolyticus* was cultured to logarithmic stage and diluted to 1 × 10^4^ cfu/μL with PBS buffer. The number of bacteria was counted by blood cell counting plate under a microscope. A total of 72 shrimp were divided equally into two groups, the *V. parahaemolyticus* infection (*Vph)* group and the control group. The immune challenge experiment was performed according to a previous study [16]. Each shrimp in the *Vph* group was injected with 10 μL *V. parahaemolyticus* with the concentration of 2 × 10^4^ cfu/μL, and that in the control group was injected with equal volume of PBS buffer. Hepatopancreas were collected at 3 h, 6 h, 12 h and 24 h after injection, nine individuals were collected at each time point from each treatment to prepare three replicates. All the samples were stored at −80 °C before total RNA extraction.

### 4.2. Total RNA Extraction and cDNA Synthesis

The samples were homogenized under liquid nitrogen with an acetabulum and pestle before total RNA extraction. Total RNA was extracted using RNAisoPlus (TaKaRa, Kyoto, Japan) according to the provided protocol. The quality of the extracted RNA was verified by 1% agarose gel electrophoresis. At the same time, the RNA quality and quantity of each sample were determined by Nanodrop 2000 (Thermo Fisher Scientific, Waltham, MA, USA). About 1 μg of total RNA from each sample was treated with PrimeScript™ RT reagen Kit with gDNA Eraser (TaKaRa, Kyoto, Japan) to remove genomic DNA contamination and synthesize cDNA following the manufacturer’s protocol.

### 4.3. Real-Time Quantitative PCR (qRT-PCR)

The expression level of *LvCrustinVII* in different samples was detected by qRT-PCR. Gene-specific primers were designed and synthesized based on *LvCrustinVII* gene sequence [30]. The 18S rRNA gene was used as the internal control. The qRT-PCR performed as following program: 95 °C for 1 min, followed by 40 cycles of 95 °C for 15 s, annealing temperature for 15 s, 72 °C for 30 s. The melting-curve analysis was added to each PCR reaction to verify the specificity of the product. Each reaction was confirmed by qPCR analysis repeated three times. Finally, the relative expression levels of *LvCrustinVII* were calculated by 2^−ΔΔCT^ method [54]. The nucleotide sequences of each primer were listed in Table 2.

### 4.4. Recombinant Protein Expression and Purification

The mature peptide region of *LvCrustinVII* was amplified using primers *rLvCrustinVII*-F/R (Table 2) with *BamH* I and *Hind* III restriction enzyme sites and purified using the MiniBEST DNA Fragment Purification Kit (TaKaRa, Kyoto, Japan) following the instruction of manufacturer. The purified PCR product and pET28a vector were digested by restriction enzymes *BamH* I and *Hind* III (TaKaRa, Kyoto, Japan). Recombinant pET28a-*LvCrustinVII* was generated by ligating the target gene fragment into a linearized vector using In-Fusion HD Cloning Kit (TaKaRa, Kyoto, Japan) and transformed into Trans5α Competent Cell (TransGen, Beijing, China). Positive clones were screened by PCR with T7-F/R primers (Table 2) and sequenced to ensure that the inserted sequence was correct. The recombinant plasmid was transformed into BL21 (DE3) Competent Cell (TransGen, Beijing, China) and cultured at 37 °C, 220 rpm. When the OD600 value reached 0.4-0.6, isopropyl-β-D-thio-galactoside (IPTG) was added with the final concentration of 1 mM. The cells were collected at 2 h, 4 h and 6 h for SDS-PAGE to detect the expression and location of target protein. Then, according to the above method, the recombinant protein was induced to the corresponding time point, and cells were collected and suspended in Equilibration Buffer (50 mM Tris-HCl pH8.0, 150 mM NaCl, 1 mM EDTA, 2 mM oxidized glutathione, 0.2 mM reduced glutathione, 10% glycerol, 1% glycine, and 8M urea). The cell suspension was sonicated, and inclusion bodies were collected by centrifugation at 8000× *g* for 10 min at 4 °C.

The inclusion bodies were solubilized in a denaturing solution containing 50 mM Na_3_PO_4_, 8 M urea and 30 mM NaCl, pH8.0. The rLvCrustinVII protein was purified under denaturing conditions with HisTALON Gravity Column Purification Kit (Clontech, Mountain View, CA, USA) following the manufacturer’s protocol and eluted progressively with a denaturing solution containing 45 mM imidazole. The purity of rLvCrustinVII was determined by Sodium dodecyl sulphate-Polyacrylamide gel electrophoresis (SDS-PAGE). The purified protein was treated with dialysis buffer (50 mM Tris-HCl pH8.0, 150 mM NaCl, 1 mM EDTA, 2 mM oxidized glutathione, 0.2 mM reduced glutathione, 10% glycerol, 1% glycine, and urea) by gradually decreasing urea from 6 M, 4 M, 2 M to 0 M. At the final step, the purified protein was dialyzed in 50 mM Tris-HCl buffer (pH 8.0), and the remaining precipitant was removed by centrifugation. The protein was collected with an ultrafiltration concentrator (Amicon, Darmstadt, Germany). The pET28a empty vector expression protein was obtained in the same way. The concentration of purified rLvCrustinVII was determined by bicinchoninic acid (BCA) protein assay (Vazyme, Nanjing, China) following the instruction of manufacturer.

### 4.5. Minimal Inhibitory Concentration (MIC) Assay

The *rLvCrustinVII* and pET28a empty vector expressed protein were diluted to 20 μM, 10 μM, 5 μM, 2.5 μM, 1.25 μM, 0.625 μM, and 0.3125 μM. *V. parahaemolyticus* and *Vibrio harveyi* were cultured in Tryptic Soy Broth (TSB) + 2% NaCl medium at 30 °C, and *Staphylococcus aureus* and *E. coli* were cultured in Lysogenic broth (LB) medium at 37 °C. All the tested bacteria were cultured to logarithmic stage and counted with a blood cell counting plate. 50 μL recombinant proteins with density gradient and 50 μL bacteria with the concentration of 2 × 10^4^ cfu/mL were added in each well of 96-well microtiter plates. A negative control without recombinant protein was also added. The plates were incubated with a horizontal shaker at room temperature for 2 h. After incubation, 150 μL corresponding medium was added and cultured for 12 h according to different strains at appropriate temperature. The absorbance of Gram-positive and Gram-negative bacteria solutions at 600 nm and 560 nm was measured using a precision micrometer (TECAN infinite M200PRO, Salzburg, Austria). The experiment was repeated three times. The concentration at which bacterial growth reached 90% of the untreated bacteria was regarded as the minimal inhibitory concentration [22]. All values were averaged using three independent measurements.

### 4.6. Microorganism Binding Assay

Western Blot assay was used to detect the microorganism binding activity of peptides following the method described previously [56,57]. In short, *V. parahaemolyticus*, *V. harveyi*, *S. aureus*, and *E. coli* were cultured at TSB or LB medium to logarithmic growth phase, respectively. Then they were fixed with 37% formaldehyde and gently shaken at 37 °C for 1 h to destroy the protease activity of microorganisms. The bacteria were collected and suspended with 450 μL PBS. 50 μL recombinant protein and a final concentration of 40 μg/mL was added to the bacterial suspension; the bacteria were gently shaken and incubated at 4 °C for 30 min. Then, the microorganism was centrifuged. The bacterial suspension was separated, and the eluent was collected. The bacteria precipitates were washed with 1 mL PBS for three times, and the washing solution was collected. Finally, the bacteria were re-suspended with PBS.

The eluent, washing solution and bacterial suspension were prepared for SDS-PAGE. Then, the protein was transferred onto a poly-vinylidene fluoride (PVDF) membrane with transfer buffer (25 mM Tris-HCl, 20 mM glycine, 0.037% SDS, and 20% ethanol) and sealed with 5% skim milk which dissolved in TBS tween (TBST) buffer (TBS buffer with 0.1% tween-20) for 2 h. The membrane was incubated with Mouse anti-His-Tag mAb (ABclonal, Wuhan, China) overnight at 4 °C and washed with TBST for three times. HRP goat anti-mouse IgG (ABclonal, Wuhan, China) was used as the secondary antibody and incubated at room temperature for 1 h. After washing with TBST buffer, rLvCrustinVII was observed using enhanced chemiluminescence (ECL) following the manufacturer’s protocol.

### 4.7. The Polysaccharides Binding Assay

The binding activity of rLvCrustinVII to several microbial polysaccharides was examined by enzyme linked immune sorbent assay (ELISA) [58]. Microbial polysaccharides, including PGN (Sigma-Aldrich, Saint Louis, MO, USA), Glu (Sigma-Aldrich, Saint Louis, MI, USA) and LPS (Sigma-Aldrich, Saint Louis, MO, USA), were used in this study. Each 96-well microtiter plate was coated with 20 µg polysaccharides (100 µL, 200 µg/mL) overnight at 4 °C. Afterwards the plate was washed three times with PBST (TBS containing 0.05%Tween20). Each well was sealed with BSA (200 µL, 3mg/mL) at 37 °C for 1 h and washed with TBST for three times. Subsequently, a series of 50 µL diluted recombinant protein were added. After incubation at room temperature for 3 h, the plate was washed with TBST for three times and incubated with Mouse anti-His-Tag mAb (diluted 1:1000 in TBS containing 0.1mg/mL BSA) at 37 °C for 1 h. The plate was washed with TBST for three times, and goat-anti-mouse Ig-HRP conjugate was added (diluted in TBS containing 0.1 mg/mL BSA at 1:1000) and incubated at 37 °C for 1 h. The plate was washed with TBST again for three times, added with 100 µL TMB (3,3,5,5-tetramethylbenzidine) to avoid light color for 5–30 min, then added with 50 µL 2 M H_2_SO_4_ to terminate the reaction. The absorbance at 450 nm was read using a precision micrometer (TECAN infinite M200PRO, Salzburg, Austria). All tests were repeated three times.

### 4.8. Microorganism Agglutination Assay

The agglutination assay was carried out according to previous reports with some modifications [58]. *V. parahaemolyticus* was cultured in TSB medium to logarithmic growth phase and harvested by centrifugation at 2000× *g*, 4 °C for 10 min. The pellets were washed three times with sterilized PBS and re-suspended in PBS at a density of 10^8^ cells mL^−1^. *V. parahaemolyticus* was labeled with fluorescein isothiocyanate (FITC) and slowly shaken overnight in the dark. Subsequently, the FITC-labeled bacteria were rinsed with PBS until there was no visible dissociated FITC. The bacteria were suspended in PBS at a density of 1 × 10^8^ cells mL^−1^. In the presence or absence of 10 mM Ca^2+^, 25 μL microbial suspension was added to 25 µL rLvCrustinVII (final concentration 50 µg/mL). Tris-HCl buffer and pET28a empty vector protein with His label were used as blank and negative controls, respectively. The mixture was incubated at room temperature for about 1 h, and 5 μL of cells were taken out and observed by fluorescence microscope. All tests were repeated three times.

### 4.9. Phagocytosis Assay

Before the experiment, *V. parahaemolyticus* was labeled with FITC according to Section 4.8. Hemolymph from healthy *L. vannamei*, centrifuged at 4 °C, 800× *g* for 20 min and resuspended with five times volume of anticoagulant. Then hemolymph was incubated with rLvCrustinVII or Tris-HCl/BSA in the presence or absence of 10 mM Ca^2+^ at 24 °C for 30 min. This mixture was added to the FITC-labeled bacteria and incubated with slow shaking at 24 °C for 45 min. Hemocytes phagocytosis was examined by flow cytometry (FACSAria, BD biosciences, Franklin Lakes, NJ, USA) and the phagocytotic rate was defined as the hemocytes ingesting bacteria/all hemocytes tested × 100%. All tests were repeated three times.

### 4.10. Ethical Statement

The present study used shrimp as experimental animals, which are not endangered invertebrates. In addition, there was no genetically modified organism used in the study. According to the national regulation (Fisheries Law of China), no permission was required to collect the animals and no formal ethics approval was required for this study.

## 5. Conclusions

In conclusion, the type VII crustin, LvCrustinVII from *L. vannamei*, was widely expressed in tested tissues, and its expression was significantly up-regulated after *V. parahaemolyticus* infection. The recombinant LvCrustinVII exhibited bacterial inhibitory activity, microorganism binding activity and polysaccharides binding activity. In addition, LvCrustinVII could agglutinate *V. parahaemolyticus* and greatly enhance phagocytosis ability of shrimp hemocytes in the presence of Ca^2+^. These results suggest that LvCrustinVII plays an important role in shrimp immunity, exerting dual functions as antimicrobial peptide and opsonin. The present data provided new insights into the multiple functions and application of crustins in the immunity of shrimp.

## Figures and Tables

**Figure 1 marinedrugs-20-00157-f001:**
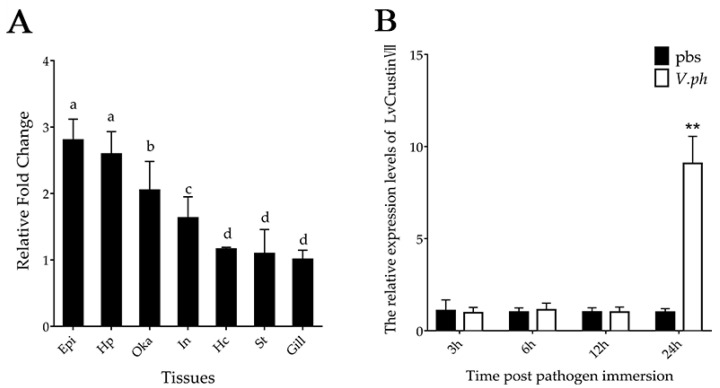
Expression profiles of *LvCrustinVII* in different tissues (**A**) and in hepatopancreas after *V. parahaemolyticus* infection (**B**). Epi, epidermis; Hp, hepatopancreas; Oka, lymphoid organ; Int, intestine; Hc, hemocytes; St, stomach; Gill, gill. Vertical bars represented the mean ± S.D. (*n* = 3). Different lowercase letters “a”, “b”, “c”, “d” in (**A**) represented significant differences among tissues at *p* < 0.05. Stars (**) in (**B**) represented significant difference between treatments at *p* < 0.01.

**Figure 2 marinedrugs-20-00157-f002:**
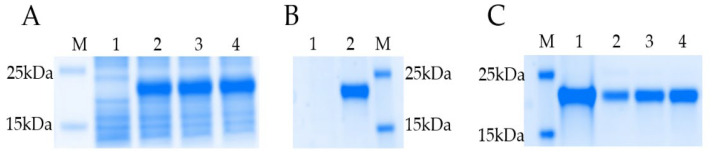
SDS-PAGE analysis of rLvCrustinVII. (**A**) Recombinant expression of rLvCrustinVII at each time point after IPTG induction. Lane M, protein Marker; Lane 1, total protein of *E. coli* before IPTG induction; Lane 2, 3 and 4, total protein after IPTG induction for 2 h, 4 h and 6 h. (**B**) Analysis of the expression form of rLvCrustinVII. Lane M, the protein Marker; Lane 1, the supernatant of *E. coli* lysis after IPTG induction for 4 h; Lane 2, the inclusion body of *E. coli* lysis after IPTG induction for 4 h. (**C**) Purification and renaturation of rLvCrustinVII. Lane M, the protein Marker; Lane 1, the total protein of induced *E. coli*; Lane 2, the refolded rLvCrustinVII; Lane 3, the concentrated rLvCrustinVII; Lane 4, the purified rLvCrustinVII.

**Figure 3 marinedrugs-20-00157-f003:**
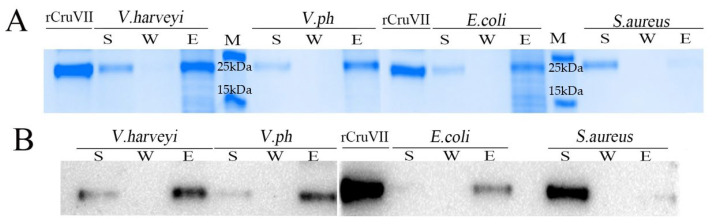
Microorganisms binding analysis of rLvCrustinVII by SDS-PAGE (**A**) and Western blot (**B**). The recombinant protein rLvCrustinVII was incubated with different formaldehyde-fixed microorganisms at 4 °C for 30 min. After incubation, the supernatants were separated by centrifugation. The pellets were washed with PBS buffer and the bound proteins were eluted with SDS-PAGE sample loading buffer. The supernatants (S), washed (W) and eluted (E) fractions were examined by SDS-PAGE and Western Blot.

**Figure 4 marinedrugs-20-00157-f004:**
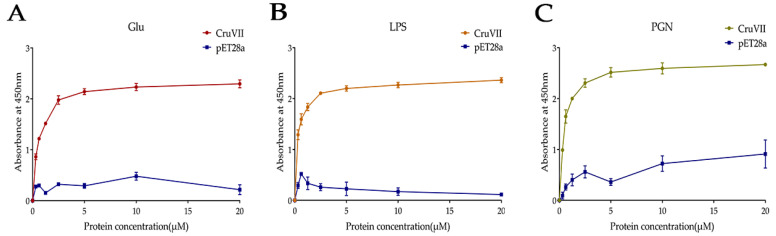
Binding activity of rLvCrustinVII to microbial polysaccharides β-1,3-Glucan, Glu (**A**), lipopolysaccharides, LPS (**B**) and peptidoglycan, PGN (**C**). The pET28a empty vector expressed protein was used as the negative control. Three independent repeats were performed, and the results are expressed as the mean ± SD.

**Figure 5 marinedrugs-20-00157-f005:**
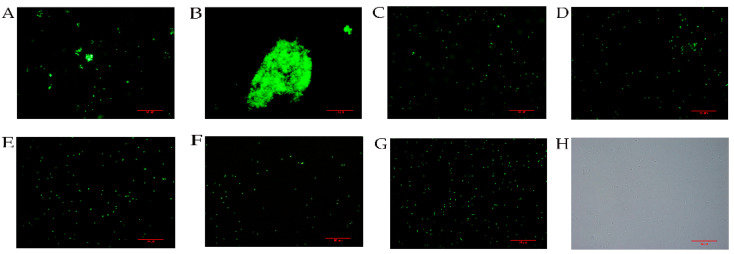
Agglutination analysis of rLvCrustinVII. FITC-labeled *V. parahaemolyticus* was incubated with rLvCrustinVII (**A**), rLvCrustinVII + Ca^2+^ (**B**), pET28a empty vector expressed protein (**C**), pET28a empty vector expressed protein + Ca^2+^ (**D**), Tris-HCl (**E**), Tris-HCl + Ca^2+^ (**F**), *V. parahaemolyticus* in fluorescence light (**G**), *V. parahaemolyticus* in visible light (**H**) for 1 h. Tris-HCl and pET28a empty vector expressed protein were used as blank and negative control, respectively. Agglutination was tested under a fluorescence microscope. Scale bar is 50 µm.

**Figure 6 marinedrugs-20-00157-f006:**
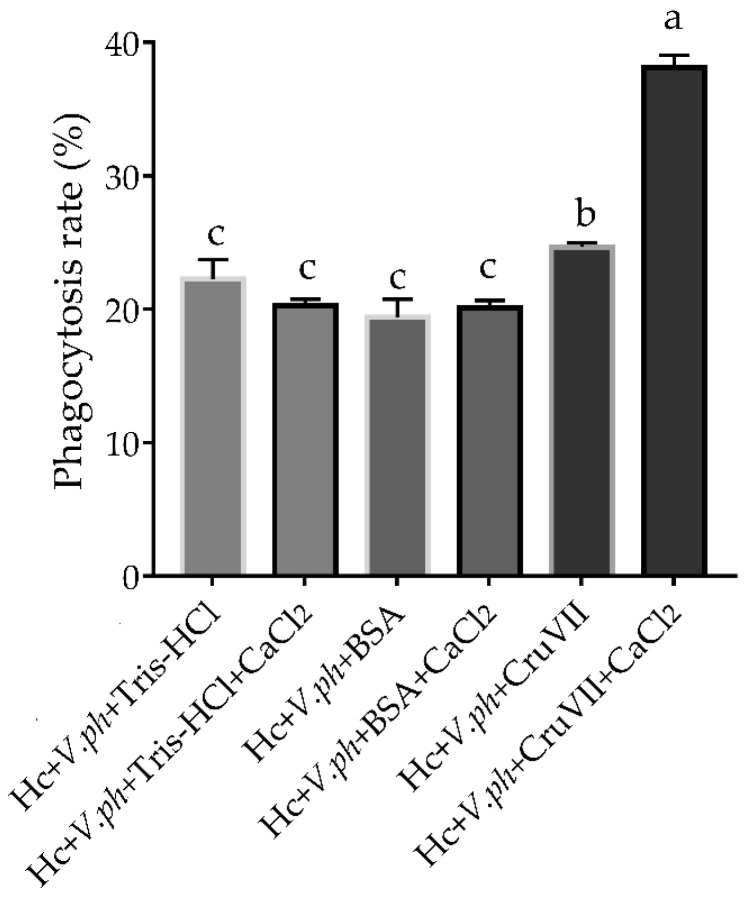
Phagocytosis analysis of shrimp hemocytes treated with rLvCrustinVII. The shrimp hemocytes suspension was incubated with rLvCrustinVII or Tris-HCl, BSA for 30 min in the absence or presence of Ca^2+^. FITC labeled *V. parahaemolyticus* was then added into the mixture and rotated slightly in dark at 18 °C for 45 min. The phagocytic rate was examined by flow cytometry. Vertical bars represented the mean ± S.D. (*n* = 3). The phagocytotic rates with significant differences (*p* < 0.05) among treatments were shown with different lowercase letters “a”, “b” and “c”.

**Table 1 marinedrugs-20-00157-t001:** Minimal bacterial growth inhibitory concentrations of *rLvCrustinVII*.

Microorganism	Minimal Inhibitory Concentrations (μM)
*V. harveyi*	2.5
*V. parahaemolyticus*	2.5
*E. coli*	>20
*S. aureus*	>20

**Table 2 marinedrugs-20-00157-t002:** Primer sequences and corresponding annealing temperature of genes.

Primer Name	Primer Sequence (5′-3′)	Annealing Temperature (°C)
18S-qF	TATACGCTAGTGGAGCTGGAA	55
18S-qR	GGGGAGGTAGTGACGAAAAAT	
LvCrustinVII-qF	CGTCCTCATCGGGCTCCTT	60
LvCrustinVII-qR	CGGCAATGTAGGCTTGGTGG	
rLvCrustinVII-F	GGATCCGAAGAATCGGAGGAAAACACGCGT	60
rLvCrustinVII-R	AAGCTTCTAGGAGGAATAGCACGGTTGCGC	
T7-F	TAATACGACTCACTATAGGG	55
T7-R	GCTAGTTATTGCTCAGCGGT	

Note: The nucleotide sequences of T7-F and T7-R were designed based on a previous study [55].

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
