# Peer review of "Characterization of the Dual Functions of LvCrustinVII from Litopenaeus vannamei as Antimicrobial Peptide and Opsonin"

_marinedrugs, 2022, doi:10.3390/md20030157_

Round 1
Reviewer 1 Report
The article concerns the characterization of the dual functions of LvCrustinVII, antimicrobial peptide from the crustacean Litopenaeus vannamei, having dual function both as antimicrobial peptide and opsonin enhancing phagocytosis of hepatocytes. The authors earlier identified LvCrustinVII but its immune functions were unclear. They continued the investigation of immune and antimicrobial properties of the recombinant peptide r-LvCrustinVII. The peptide revealed strong antimicrobial action on Gram-negative bacteria Vibrio harveyi and V. parahaemolyticus, but very low activity against Gram-positive bacteria Staphylococcus aureus. rLvCrustinVII binds strongly to V. harveyi and V. parahaemolyticus, as well as their cell wall components. rLvCrustinVII effectively agglutinates V. parahaemolyticus and enhances hemocyte phagocytosis especially in the presence of calcium ions. The obtained results clear illustrate the peculiarity of the immune properties of rLvCrustinVII. The methodical level of the work is high, all the methods are adequately used, the article is very well written.
Only few very minor corrections should be done. The name of crustacean-producer of the antimicrobial peptide ahouls be inserted into the title such as: Characterization of the dual functions of LvCrustinVII from the crustacean Litopenaeus vannamei as antimicrobial peptide and opsonin” or by any similar way.
The spaces between the figures and units should be inserted in the Experimental and uL should be replaced with µL (Line 343). Minor correction of English in the Experimental is necessary.
The article may be published after very minor corrections.
Author Response
Comment (C) 1:The name of crustacean-producer of the antimicrobial peptide should be inserted into the title such as: Characterization of the dual functions of LvCrustinVII from the crustacean Litopenaeus vannamei as antimicrobial peptide and opsonin” or by any similar way.
Response (R) 1:The title of the manuscript has been changed to: Characterization of the dual functions of LvCrustinVII from Litopenaeus vannamei as antimicrobial peptide and opsonin.
C2: The spaces between the figures and units should be inserted in the Experimental and uL should be replaced with µL (Line 343).
R2: The spaces between the figures and units have been inserted in the Experimental, and uL has been replaced with µL.
Reviewer 2 Report
In the present manuscript, Hu and co-workers, highlighted the immune function of type VII crustin, LvCrustinVII, by studying its expression in Litopenaeus vannamei`s tissues. Furthermore, the authors proved the antimicrobial activity of the recombinant LvCrustinVII against Gram-negative bacteria Vibrio harveyi and V. parahaemolyticus using also the binding assay and agglutination tests.
The article is well written and structured. The results obtained are very well presented and interpreted. I only have to mention that:
- It would be better to specify the concentration of parahaemolyticus in cfu/mL (or cfu/uL, or cfu/sample, etc.) and not just cfu (line 260)
- The same comment for the line 326 from the MIC assay.
- Why you chose the concentration of 2×105 cfu of V. parahaemolyticus in the experiment for the gene expression? Please, give a reference.
- Please check again, and be more specific when you present the MIC value. Usually, the MIC is expressed like a unique value and not like a range as it was written (1.25~2.5µM, line 134).
- The nucleotide listed in the table 2 are original or they are obtained from the literature? If it is the case, please give the references.
- Which predictor for the molecular mass was used? (line 119)
- Delete one point from the” (Figure1B)..” (Line 103)
- Please, check if it is the case to write in the manuscript, the Ethical Statement
Author Response
Comment (C) 1: It would be better to specify the concentration of parahaemolyticus in cfu/mL (or cfu/uL, or cfu/sample, etc.) and not just cfu (line 260). The same comment for the line 326 from the MIC assay.
Response (R) 1: Thanks for the suggestion. The concentration of parahaemolyticus has been added in the revised manuscript.
C2: Why you chose the concentration of 2×105 cfu of V. parahaemolyticus in the experiment for the gene expression? Please, give a reference.
R2: The concentration was used according to a previous study. The reference was added in the revised manuscript.
C3: Please check again, and be more specific when you present the MIC value. Usually, the MIC is expressed like a unique value and not like a range as it was written (1.25~2.5µM, line 134).
R3: Thanks for the suggestion. The MIC values have been modified in the revised manuscript.
C4: The nucleotide listed in the table 2 are original or they are obtained from the literature? If it is the case, please give the references.
R4: The nucleotide sequences of T7-F and T7-R were cited from a precious study, which has been added in the revised manuscript.
C5: Which predictor for the molecular mass was used? (line 119)
R5: The molecular mass was predictor by the ExPASy (https://web.expasy.org/protparam/). The information has been added in the revised manuscript.
C6: Delete one point from the” (Figure1B)..” (Line 103)
R6: It has been deleted.
C7: Please, check if it is the case to write in the manuscript, the Ethical Statement
R7: Thanks for the suggestion. The Ethical Statement has been added.
Reviewer 3 Report
This study explores functional aspects of a crustacean-derived protein, which was produced by heterologous expression. The manuscript needs to be carefully reviewed before publication.
- The manuscript needs to be revised. There is excessive repetition of some words and sentences. For example "diseases" appear three times in two sentences in a row (lines 36-38). Revise aquaculture, antibiotics, pathogens, granules (Introduction section) and so on. Revise lines 29-30 and lines 405-406.
- Figure 5 should include a microscopic image in visible light to show the bacteria in standard preparation. Protein concentration should be specified in this section for better understanding. Why did the authors use a much higher concentration than the MIC? At lower concentrations is this phenomenon observed?
- Please review this statement: "...The type VII crustin LvCrustinVII has strong inhibitory activity against Gram-negative while weak activity against a tested Gram-positive bacterium, which would provide a new target for drug development. .." This information seems to contradict the findings presented in Table 1. Differences were not observed for antibacterial activity against S. aureus and E. coli.
- Line 61-62: AMPs produced by the innate immunity in crustaceans present some novel drug properties. Which?
- What is the primary structure of LvCrustinVII? Does the mass determined in the gel agree with the sequence? According to the result presented in the gel, the most appropriate term would be protein and not peptide as described in the title and other sections of manuscript.
- Molecular weights of the marker used in Figure 3A must be identified.
- The discussion contains only 3 paragraphs and should be enriched.
- Protein concentration used in binding assays must be specified in the results. The authors should justify the chosen concentration.
- Line 322: Concentrations are usually presented in ascending order.
Author Response
Comment (C) 1: The manuscript needs to be revised. There is excessive repetition of some words and sentences. For example, "diseases" appear three times in two sentences in a row (lines 36-38). Revise aquaculture, antibiotics, pathogens, granules (Introduction section) and so on. Revise lines 29-30 and lines 405-406.
Response (R) 1: Thanks for the suggestion. The manuscript has been carefully modified to avoid repetition of some words and sentences.
C2: Figure 5 should include a microscopic image in visible light to show the bacteria in standard preparation. Protein concentration should be specified in this section for better understanding. Why did the authors use a much higher concentration than the MIC? At lower concentrations is this phenomenon observed?
R2: Microscopic images in fluorescence light and in visible light to show the bacteria in standard preparation have been added in the revised Figure 5. The protein final concentration was 50 µg/mL in binding assays, with the molar concentration of 2.7 µM, which approximately similar to the MIC (2.5µM).
C3: Please review this statement: "...The type VII crustin LvCrustinVII has strong inhibitory activity against Gram-negative while weak activity against a tested Gram-positive bacterium, which would provide a new target for drug development..." This information seems to contradict the findings presented in Table 1. Differences were not observed for antibacterial activity against S. aureus and E. coli.
R: The sentence was confused. The present study found that LvCrustinVII mainly exhibited strong activity against Vibrios rather than other bacteria. We have modified this sentence in the revised manuscript.
C4: Line 61-62: AMPs produced by the innate immunity in crustaceans present some novel drug properties. Which?
R4: The description of crustacean AMPs and their drug properties have been specified in the revised manuscript.
C5: What is the primary structure of LvCrustinVII? Does the mass determined in the gel agree with the sequence? According to the result presented in the gel, the most appropriate term would be protein and not peptide as described in the title and other sections of manuscript.
R5: The primary structure of LvCrustinVII was reported in one of our previous articles (reference 29). In the present study, the recombinant rLvCrustinVII in the gel had an identical mass to the predicted molecular mass, as described in the Results (the last sentence in 2.2). Although it is a protein with much larger mass than conventional peptides, this kind of proteins are traditionally termed as antimicrobial peptides. Therefore, we used peptide rather than protein in the main text.
C6: Molecular weights of the marker used in Figure 3A must be identified.
R6: Molecular weights of the marker has been added in the revised manuscript.
C7: The discussion contains only 3 paragraphs and should be enriched.
R7: Thanks for the suggestion. The discussion has been enriched in the revised manuscript.
C8: Protein concentration used in binding assays must be specified in the results. The authors should justify the chosen concentration.
R8: The protein final concentration was 40 µg/mL in binding assays, with the molar concentration of 2.5 µM, which was the same as the MIC.
C9: Line 322: Concentrations are usually presented in ascending order.
R9: Since the recombinant protein is diluted step by step from the highest concentration, the concentration was shown in descending order.
Round 2
Reviewer 3 Report
I recommend the publication of this manuscript. The authors have addressed most of my concerns.